# Cooperative Inverse Reinforcement Learning

**Dylan Hadfield-Menell**[*]   **Anca Dragan**   **Pieter Abbeel**   **Stuart Russell**

Electrical Engineering and Computer Science
University of California at Berkeley
Berkeley, CA 94709

## Abstract

For an autonomous system to be helpful to humans and to pose no unwarranted risks, it needs to align its values with those of the humans in its environment in such a way that its actions contribute to the maximization of value for the humans. We propose a formal definition of the value alignment problem as *cooperative inverse reinforcement learning* (CIRL). A CIRL problem is a cooperative, partial-information game with two agents, human and robot; both are rewarded according to the human's reward function, but the robot does not initially know what this is. In contrast to classical IRL, where the human is assumed to act optimally in isolation, optimal CIRL solutions produce behaviors such as active teaching, active learning, and communicative actions that are more effective in achieving value alignment. We show that computing optimal joint policies in CIRL games can be reduced to solving a POMDP, prove that optimality in isolation is suboptimal in CIRL, and derive an approximate CIRL algorithm.

## 1   Introduction

"*If we use, to achieve our purposes, a mechanical agency with whose operation we cannot interfere effectively . . . we had better be quite sure that the purpose put into the machine is the purpose which we really desire.*" So wrote Norbert Wiener (1960) in one of the earliest explanations of the problems that arise when a powerful autonomous system operates with an incorrect objective. This *value alignment* problem is far from trivial. Humans are prone to mis-stating their objectives, which can lead to unexpected implementations. In the myth of King Midas, the main character learns that wishing for 'everything he touches to turn to gold' leads to disaster. In a reinforcement learning context, Russell & Norvig (2010) describe a seemingly reasonable, but incorrect, reward function for a vacuum robot: if we reward the action of cleaning up dirt, the optimal policy causes the robot to repeatedly dump and clean up the same dirt.

A solution to the value alignment problem has long-term implications for the future of AI and its relationship to humanity (Bostrom, 2014) and short-term utility for the design of usable AI systems. Giving robots the right objectives and enabling them to make the right trade-offs is crucial for self-driving cars, personal assistants, and human–robot interaction more broadly.

The field of *inverse reinforcement learning* or IRL (Russell, 1998; Ng & Russell, 2000; Abbeel & Ng, 2004) is certainly relevant to the value alignment problem. An IRL algorithm infers the reward function of an agent from observations of the agent's behavior, which is assumed to be optimal (or approximately so). One might imagine that IRL provides a simple solution to the value alignment problem: the robot observes human behavior, learns the human reward function, and behaves according to that function. This simple idea has two flaws. The first flaw is obvious: we don't want the robot to adopt the human reward function as its own. For example, human behavior (especially in the morning) often conveys a desire for coffee, and the robot can learn this with IRL, but we don't want the robot to want coffee! This flaw is easily fixed: we need to formulate the value

---

[*] {dhm, anca, pabbeel, russell}@cs.berkeley.edu

alignment problem so that the robot always has the fixed objective of optimizing reward *for the human*, and becomes better able to do so as it learns what the human reward function is.

The second flaw is less obvious, and less easy to fix. IRL assumes that observed behavior is optimal in the sense that it accomplishes a given task efficiently. This precludes a variety of useful teaching behaviors. For example, efficiently making a cup of coffee, while the robot is a passive observer, is a *inefficient* way to teach a robot to get coffee. Instead, the human should perhaps *explain* the steps in coffee preparation and *show* the robot where the backup coffee supplies are kept and what do if the coffee pot is left on the heating plate too long, while the robot might *ask* what the button with the puffy steam symbol is for and *try its hand* at coffee making with guidance from the human, even if the first results are undrinkable. None of these things fit in with the standard IRL framework.

**Cooperative inverse reinforcement learning.** We propose, therefore, that value alignment should be formulated as a *cooperative* and *interactive* reward maximization process. More precisely, we define a *cooperative inverse reinforcement learning* (CIRL) game as a two-player game of partial information, in which the "human", $\mathbf{H}$, knows the reward function (represented by a generalized parameter $\theta$), while the "robot", $\mathbf{R}$, does not; the robot's payoff is exactly the human's actual reward. Optimal solutions to this game maximize human reward; we show that solutions may involve active instruction by the human and active learning by the robot.

**Reduction to POMDP and Sufficient Statistics.** As one might expect, the structure of CIRL games is such that they admit more efficient solution algorithms than are possible for general partial-information games. Let $(\pi^{\mathbf{H}}, \pi^{\mathbf{R}})$ be a pair of policies for human and robot, each depending, in general, on the complete history of observations and actions. A policy pair yields an expected sum of rewards for each player. CIRL games are cooperative, so there is a well-defined optimal policy pair that maximizes value.[2] In Section 3 we reduce the problem of computing an optimal policy pair to the solution of a (single-agent) POMDP. This shows that the robot's posterior over $\theta$ is a sufficient statistic, in the sense that there are optimal policy pairs in which the robot's behavior depends only on this statistic. Moreover, the complexity of solving the POMDP is exponentially lower than the NEXP-hard bound that (Bernstein et al., 2000) obtained by reducing a CIRL game to a general DEC-POMDP.

**Apprenticeship Learning and Suboptimality of IRL-Like Solutions.** In Section 3.3 we model apprenticeship learning (Abbeel & Ng, 2004) as a two-phase CIRL game. In the first phase, the learning phase, both $\mathbf{H}$ and $\mathbf{R}$ can take actions and this lets $\mathbf{R}$ learn about $\theta$. In the second phase, the deployment phase, $\mathbf{R}$ uses what it learned to maximize reward (without supervision from $\mathbf{H}$). We show that classic IRL falls out as the best-response policy for $\mathbf{R}$ under the assumption that the human's policy is "demonstration by expert" (DBE), i.e., acting optimally *in isolation* as if no robot exists. But we show also that this DBE/IRL policy pair is not, in general, optimal: even if the robot expects expert behavior, demonstrating expert behavior is not the best way to teach that algorithm.

We give an algorithm that approximately computes $\mathbf{H}$'s best response when $\mathbf{R}$ is running IRL under the assumption that rewards are linear in $\theta$ and state features. Section 4 compares this best-response policy with the DBE policy in an example game and provides empirical confirmation that the best-response policy, which turns out to "teach" $\mathbf{R}$ about the value landscape of the problem, is better than DBE. Thus, designers of apprenticeship learning systems should *expect* that users will violate the assumption of expert demonstrations in order to better communicate information about the objective.

## 2 Related Work

Our proposed model shares aspects with a variety of existing models. We divide the related work into three categories: inverse reinforcement learning, optimal teaching, and principal–agent models.

**Inverse Reinforcement Learning.** Ng & Russell (2000) define *inverse reinforcement learning* (IRL) as follows: "**Given** measurements of an [actor]'s behavior over time. ... **Determine** the reward function being optimized." The key assumption IRL makes is that the observed behavior is optimal in the sense that the observed trajectory maximizes the sum of rewards. We call this the *demonstration-by-expert* (DBE) assumption. One of our contributions is to prove that this may be *suboptimal* behavior in a CIRL game, as $\mathbf{H}$ may choose to accept less reward on a particular action in order to *convey more information* to $\mathbf{R}$. In CIRL the DBE assumption prescribes a fixed policy

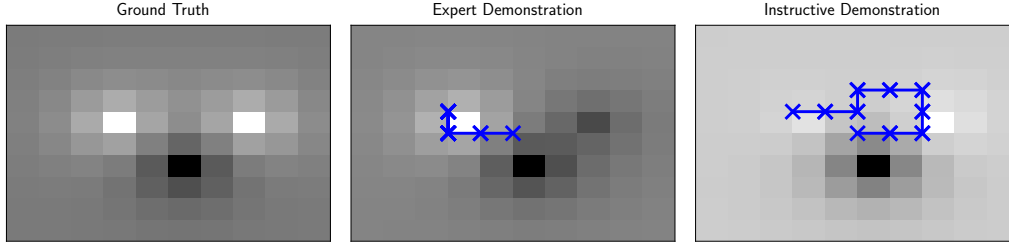

Figure 1: The difference between demonstration-by-expert and instructive demonstration in the mobile robot navigation problem from Section 4. Left: The ground truth reward function. Lighter grid cells indicates areas of higher reward. Middle: The demonstration trajectory generated by the expert policy, superimposed on the maximum a-posteriori reward function the robot infers. The robot successfully learns where the maximum reward is, but little else. Right: An instructive demonstration generated by the algorithm in Section 3.4 superimposed on the maximum a-posteriori reward function that the robot infers. This demonstration highlights both points of high reward and so the robot learns a better estimate of the reward.

for **H**. As a result, many IRL algorithms can be derived as state estimation for a best response to different $\pi^{\mathbf{H}}$, where the state includes the unobserved reward parametrization $\theta$.

Ng & Russell (2000), Abbeel & Ng (2004), and Ratliff et al. (2006) compute constraints that characterize the set of reward functions so that the observed behavior maximizes reward. In general, there will be many reward functions consistent with this constraint. They use a max-margin heuristic to select a single reward function from this set as their estimate. In CIRL, the constraints they compute characterize **R**'s belief about $\theta$ under the DBE assumption.

Ramachandran & Amir (2007) and Ziebart et al. (2008) consider the case where $\pi^{\mathbf{H}}$ is "noisily expert," i.e., $\pi^{\mathbf{H}}$ is a Boltzmann distribution where actions or trajectories are selected in proportion to the exponent of their value. Ramachandran & Amir (2007) adopt a Bayesian approach and place an explicit prior on rewards. Ziebart et al. (2008) places a prior on reward functions indirectly by assuming a uniform prior over trajectories. In our model, these assumptions are variations of DBE and both implement state estimation for a best response to the appropriate fixed **H**.

Natarajan et al. (2010) introduce an extension to IRL where **R** observes multiple actors that cooperate to maximize a common reward function. This is a different type of cooperation than we consider, as the reward function is common knowledge and **R** is a passive observer. Waugh et al. (2011) and Kuleshov & Schrijvers (2015) consider the problem of inferring payoffs from observed behavior in a general (i.e., non-cooperative) game given observed behavior. It would be interesting to consider an analogous extension to CIRL, akin to mechanism design, in which **R** tries to maximize collective utility for a group of **H**s that may have competing objectives.

Fern et al. (2014) consider a *hidden-goal* MDP, a special case of a POMDP where the goal is an unobserved part of the state. This can be considered a special case of CIRL, where $\theta$ encodes a particular goal state. The frameworks share the idea that **R** helps **H**. The key difference between the models lies in the treatment of the human (the agent in their terminology). Fern et al. (2014) model the human as part of the environment. In contrast, we treat **H** as an actor in a decision problem that both actors collectively solve. This is crucial to modeling the human's incentive to *teach*.

**Optimal Teaching.** Because CIRL incentivizes the human to teach, as opposed to maximizing reward in isolation, our work is related to optimal teaching: finding examples that optimally train a learner (Balbach & Zeugmann, 2009; Goldman et al., 1993; Goldman & Kearns, 1995). The key difference is that efficient learning is the *objective* of optimal teaching, while it emerges as a *property* of optimal equilibrium behavior in CIRL.

Cakmak & Lopes (2012) consider an application of optimal teaching where the goal is to teach the learner the reward function for an MDP. The teacher gets to pick initial states from which an expert executes the reward-maximizing trajectory. The learner uses IRL to infer the reward function, and the teacher picks initial states to minimize the learner's uncertainty. In CIRL, this approach can be characterized as an approximate algorithm for **H** that greedily minimizes the entropy of **R**'s belief.

Beyond teaching, several models focus on taking actions that convey some underlying state, not necessarily a reward function. Examples include finding a motion that best communicates an agent's intention (Dragan & Srinivasa, 2013), or finding a natural language utterance that best communicates

a particular grounding (Golland et al., 2010). All of these approaches model the observer's inference process and compute actions (motion or speech) that maximize the probability an observer infers the correct hypothesis or goal. Our approximate solution to CIRL is analogous to these approaches, in that we compute actions that are informative of the correct reward function.

**Principal–agent models.** Value alignment problems are not intrinsic to artificial agents. Kerr (1975) describes a wide variety of misaligned incentives in the aptly titled "On the folly of rewarding A, while hoping for B." In economics, this is known as the principal–agent problem: the principal (e.g., the employer) specifies incentives so that an agent (e.g., the employee) maximizes the principal's profit (Jensen & Meckling, 1976).

Principal–agent models study the problem of generating appropriate incentives in a non-cooperative setting with asymmetric information. In this setting, misalignment arises because the agents that economists model are people and intrinsically have their own desires. In AI, misalignment arises entirely from the information asymmetry between the principal and the agent; if we could characterize the correct reward function, we could program it into an artificial agent. Gibbons (1998) provides a useful survey of principal–agent models and their applications.

## 3 Cooperative Inverse Reinforcement Learning

This section formulates CIRL as a two-player Markov game with identical payoffs, reduces the problem of computing an optimal policy pair for a CIRL game to solving a POMDP, and characterizes *apprenticeship learning* as a subclass of CIRL games.

### 3.1 CIRL Formulation

**Definition 1.** *A* cooperative inverse reinforcement learning *(CIRL) game $M$ is a two-player Markov game with identical payoffs between a human or principal, $\mathbf{H}$, and a robot or agent, $\mathbf{R}$. The game is described by a tuple, $M = \langle \mathcal{S}, \{\mathcal{A}^{\mathbf{H}}, \mathcal{A}^{\mathbf{R}}\}, T(\cdot|\cdot,\cdot,\cdot), \{\Theta, R(\cdot,\cdot,\cdot;\cdot)\}, P_0(\cdot,\cdot), \gamma \rangle$, with the following definitions:*

   $\mathcal{S}$  *a set of world states: $s \in \mathcal{S}$.*
   $\mathcal{A}^{\mathbf{H}}$  *a set of actions for $\mathbf{H}$: $a^{\mathbf{H}} \in \mathcal{A}^{\mathbf{H}}$.*
   $\mathcal{A}^{\mathbf{R}}$  *a set of actions for $\mathbf{R}$: $a^{\mathbf{R}} \in \mathcal{A}^{\mathbf{R}}$.*
   $T(\cdot|\cdot,\cdot,\cdot)$  *a conditional distribution on the next world state, given previous state and action for both agents: $T(s'|s, a^{\mathbf{H}}, a^{\mathbf{R}})$.*
   $\Theta$  *a set of possible static reward parameters, only observed by $\mathbf{H}$: $\theta \in \Theta$.*
   $R(\cdot,\cdot,\cdot;\cdot)$  *a parameterized reward function that maps world states, joint actions, and reward parameters to real numbers. $R : \mathcal{S} \times \mathcal{A}^{\mathbf{H}} \times \mathcal{A}^{\mathbf{R}} \times \Theta \to \mathbb{R}$.*
   $P_0(\cdot,\cdot)$  *a distribution over the initial state, represented as tuples: $P_0(s_0, \theta)$*
   $\gamma$  *a discount factor: $\gamma \in [0, 1]$.*

We write the reward for a state–parameter pair as $R(s, a^{\mathbf{H}}, a^{\mathbf{R}}; \theta)$ to distinguish the static reward parameters $\theta$ from the changing world state $s$. The game proceeds as follows. First, the initial state, a tuple $(s, \theta)$, is sampled from $P_0$. $\mathbf{H}$ observes $\theta$, but $\mathbf{R}$ does not. This observation model captures the notion that only the human knows the reward function, while both actors know a prior distribution over possible reward functions. At each timestep $t$, $\mathbf{H}$ and $\mathbf{R}$ observe the current state $s_t$ and select their actions $a_t^{\mathbf{H}}, a_t^{\mathbf{R}}$. Both actors receive reward $r_t = R(s_t, a_t^{\mathbf{H}}, a_t^{\mathbf{R}}; \theta)$ and observe each other's action selection. A state for the next timestep is sampled from the transition distribution, $s_{t+1} \sim P_T(s'|s_t, a_t^{\mathbf{H}}, a_t^{\mathbf{R}})$, and the process repeats.

Behavior in a CIRL game is defined by a pair of policies, $(\pi^{\mathbf{H}}, \pi^{\mathbf{R}})$, that determine action selection for $\mathbf{H}$ and $\mathbf{R}$ respectively. In general, these policies can be arbitrary functions of their observation histories; $\pi^{\mathbf{H}} : \left[ \mathcal{A}^{\mathbf{H}} \times \mathcal{A}^{\mathbf{R}} \times \mathcal{S} \right]^* \times \Theta \to \mathcal{A}^{\mathbf{H}}, \pi^{\mathbf{R}} : \left[ \mathcal{A}^{\mathbf{H}} \times \mathcal{A}^{\mathbf{R}} \times \mathcal{S} \right]^* \to \mathcal{A}^{\mathbf{R}}$. The optimal joint policy is the policy that maximizes *value*. The value of a state is the expected sum of discounted rewards under the initial distribution of reward parameters and world states.

**Remark 1.** *A key property of CIRL is that the human and the robot get rewards determined by the same reward function. This incentivizes the human to teach and the robot to learn without explicitly encoding these as objectives of the actors.*

## 3.2 Structural Results for Computing Optimal Policy Pairs

The analogue in CIRL to computing an optimal policy for an MDP is the problem of computing an optimal policy pair. This is a pair of policies that maximizes the expected sum of discounted rewards. This is not the same as 'solving' a CIRL game, as a real world implementation of a CIRL agent must account for coordination problems and strategic uncertainty (Boutilier, 1999). The optimal policy pair represents the best $\mathbf{H}$ and $\mathbf{R}$ can do if they can coordinate perfectly before $\mathbf{H}$ observes $\theta$. Computing an optimal joint policy for a cooperative game is the solution to a *decentralized-partially observed Markov decision process* (Dec-POMDP). Unfortunately, Dec-POMDPs are NEXP-complete (Bernstein et al., 2000) so general Dec-POMDP algorithms have a computational complexity that is doubly exponential. Fortunately, CIRL games have special structure that reduces this complexity.

Nayyar et al. (2013) shows that a Dec-POMDP can be reduced to a *coordination*-POMDP. The actor in this POMDP is a coordinator that observes all common observations and specifies a policy for each actor. These policies map each actor's private information to an action. The structure of a CIRL game implies that the private information is limited to $\mathbf{H}$'s initial observation of $\theta$. This allows the reduction to a coordination-POMDP to preserve the size of the (hidden) state space, making the problem easier.

**Theorem 1.** *Let $M$ be an arbitrary CIRL game with state space $\mathcal{S}$ and reward space $\Theta$. There exists a (single-actor) POMDP $M_{\mathbf{C}}$ with (hidden) state space $\mathcal{S}_{\mathbf{C}}$ such that $|\mathcal{S}_{\mathbf{C}}| = |\mathcal{S}| \cdot |\Theta|$ and, for any policy pair in $M$, there is a policy in $M_{\mathbf{C}}$ that achieves the same sum of discounted rewards.*

Theorem proofs can be found in the supplementary material. An immediate consequence of this result is that $\mathbf{R}$'s belief about $\theta$ is a sufficient statistic for optimal behavior.

**Corollary 1.** *Let $M$ be a CIRL game. There exists an optimal policy pair $(\pi^{\mathbf{H}^*}, \pi^{\mathbf{R}^*})$ that only depends on the current state and $\mathbf{R}$'s belief.*

**Remark 2.** *In a general Dec-POMDP, the hidden state for the coordinator-POMDP includes each actor's history of observations. In CIRL, $\theta$ is the only private information so we get an exponential decrease in the complexity of the reduced problem. This allows one to apply general POMDP algorithms to compute optimal joint policies in CIRL.*

It is important to note that the reduced problem may still be very challenging. POMDPs are difficult in their own right and the reduced problem still has a much larger action space. That being said, this reduction is still useful in that it characterizes optimal joint policy computation for CIRL as significantly easier than Dec-POMDPs. Furthermore, this theorem can be used to justify approximate methods (e.g., iterated best response) that only depend on $\mathbf{R}$'s belief state.

## 3.3 Apprenticeship Learning as a Subclass of CIRL Games

A common paradigm for robot learning from humans is *apprenticeship learning*. In this paradigm, a human gives demonstrations to a robot of a sample task and the robot is asked to imitate it in a subsequent task. In what follows, we formulate apprenticeship learning as turn-based CIRL with a learning phase and a deployment phase. We characterize IRL as the best response (i.e., the policy that maximizes reward given a fixed policy for the other player) to a demonstration-by-expert policy for $\mathbf{H}$. We also show that this policy is, in general, *not part of an optimal joint policy* and so IRL is generally a suboptimal approach to apprenticeship learning.

**Definition 2.** *(ACIRL) An* apprenticeship cooperative inverse reinforcement learning *(ACIRL) game is a turn-based CIRL game with two phases: a learning phase where the human and the robot take turns acting, and a deployment phase, where the robot acts independently.*

**Example.** Consider an example apprenticeship task where $\mathbf{R}$ needs to help $\mathbf{H}$ make office supplies. $\mathbf{H}$ and $\mathbf{R}$ can make paperclips and staples and the unobserved $\theta$ describe $\mathbf{H}$'s preference for paperclips vs staples. We model the problem as an ACIRL game in which the learning and deployment phase each consist of an individual action. The world state in this problem is a tuple $(p_s, q_s, t)$ where $p_s$ and $q_s$ respectively represent the number of paperclips and staples $\mathbf{H}$ owns. $t$ is the round number. An action is a tuple $(p_a, q_a)$ that produces $p_a$ paperclips and $q_a$ staples. The human can make 2 items total: $\mathcal{A}^{\mathbf{H}} = \{(0, 2), (1, 1), (2, 0)\}$. The robot has different capabilities. It can make 50 units of each item or it can choose to make 90 of a single item: $\mathcal{A}^{\mathbf{R}} = \{(0, 90), (50, 50), (90, 0)\}$. We let $\Theta = [0, 1]$ and define $R$ so that $\theta$ indicates the relative preference between paperclips and staples:$R(s, (p_a, q_a); \theta) = \theta p_a + (1 - \theta)q_a$. $\mathbf{R}$'s action is ignored when $t = 0$ and $\mathbf{H}$'s is ignored when $t = 1$. At $t = 2$, the game is over, so the game transitions to a sink state, $(0, 0, 2)$.

**Deployment phase — maximize mean reward estimate.** It is simplest to analyze the deployment phase first. $\mathbf{R}$ is the only actor in this phase so it get no more observations of its reward. We have shown that $\mathbf{R}$'s belief about $\theta$ is a sufficient statistic for the optimal policy. This belief about $\theta$ induces a distribution over MDPs. A straightforward extension of a result due to Ramachandran & Amir (2007) shows that $\mathbf{R}$'s optimal deployment policy maximizes reward for the mean reward function.

**Theorem 2.** *Let $M$ be an ACIRL game. In the deployment phase, the optimal policy for $\mathbf{R}$ maximizes reward in the MDP induced by the mean $\theta$ from $\mathbf{R}$'s belief.*

In our example, suppose that $\pi^{\mathbf{H}}$ selects $(0, 2)$ if $\theta \in [0, \frac{1}{3})$, $(1, 1)$ if $\theta \in [\frac{1}{3}, \frac{2}{3}]$ and $(2, 0)$ otherwise. $\mathbf{R}$ begins with a uniform prior on $\theta$ so observing, e.g., $a^{\mathbf{H}} = (0, 2)$ leads to a posterior distribution that is uniform on $[0, \frac{1}{3})$. Theorem 2 shows that the optimal action maximizes reward for the mean $\theta$ so an optimal $\mathbf{R}$ behaves as though $\theta = \frac{1}{6}$ during the deployment phase.

**Learning phase — expert demonstrations are not optimal.** A wide variety of apprenticeship learning approaches assume that demonstrations are given by an expert. We say that $\mathbf{H}$ satisfies the *demonstration-by-expert* (DBE) assumption in ACIRL if she greedily maximizes immediate reward on her turn. This is an 'expert' demonstration because it demonstrates a reward maximizing action but does not account for that action's impact on $\mathbf{R}$'s belief. We let $\pi^{\mathbf{E}}$ represent the DBE policy.

Theorem 2 enables us to characterize the best response for $\mathbf{R}$ when $\pi^{\mathbf{H}} = \pi^{\mathbf{E}}$: use IRL to compute the posterior over $\theta$ during the learning phase and then act to maximize reward under the mean $\theta$ in the deployment phase. We can also analyze the DBE assumption itself. In particular, we show that $\pi^{\mathbf{E}}$ is not $\mathbf{H}$'s best response when $\pi^{\mathbf{R}}$ is a best response to $\pi^{\mathbf{E}}$.

**Theorem 3.** *There exist ACIRL games where the best-response for $\mathbf{H}$ to $\pi^{\mathbf{R}}$ violates the expert demonstrator assumption. In other words, if $\mathbf{br}(\pi)$ is the best response to $\pi$, then $\mathbf{br}(\mathbf{br}(\pi^{\mathbf{E}})) \neq \pi^{\mathbf{E}}$.*

The supplementary material proves this theorem by computing the optimal equilibrium for our example. In that equilibrium, $\mathbf{H}$ selects $(1, 1)$ if $\theta \in [\frac{41}{92}, \frac{51}{92}]$. In contrast, $\pi^{\mathbf{E}}$ only chooses $(1, 1)$ if $\theta = 0.5$. The change arises because there are situations (e.g., $\theta = 0.49$) where the immediate loss of reward to $\mathbf{H}$ is worth the improvement in $\mathbf{R}$'s estimate of $\theta$.

**Remark 3.** *We should expect experienced users of apprenticeship learning systems to present demonstrations optimized for fast learning rather than demonstrations that maximize reward.*

Crucially, the demonstrator is incentivized to deviate from $\mathbf{R}$'s assumptions. This has implications for the design and analysis of apprenticeship systems in robotics. Inaccurate assumptions about user behavior are notorious for exposing bugs in software systems (see, e.g., Leveson & Turner (1993)).

### 3.4 Generating Instructive Demonstrations

Now, we consider the problem of computing $\mathbf{H}$'s best response when $\mathbf{R}$ uses IRL as a state estimator. For our toy example, we computed solutions exhaustively, for realistic problems we need a more efficient approach. Section 3.2 shows that this can be reduced to an POMDP where the state is a tuple of world state, reward parameters, and $\mathbf{R}$'s belief. While this is easier than solving a general DEC-POMDP, it is a computational challenge. If we restrict our attention to the case of linear reward functions we can develop an efficient algorithm to compute an approximate best response.

Specifically, we consider the case where the reward for a state $(s, \theta)$ is defined as a linear combination of state features for some feature function $\phi : R(s, a^{\mathbf{H}}, a^{\mathbf{R}}; \theta) = \phi(s)^{\top}\theta$. Standard results from the IRL literature show that policies with the same expected feature counts have the same value (Abbeel & Ng, 2004). Combined with Theorem 2, this implies that the optimal $\pi^{\mathbf{R}}$ under the DBE assumption computes a policy that matches the observed feature counts from the learning phase.

This suggests a simple approximation scheme. To compute a demonstration trajectory $\tau^{\mathbf{H}}$, first compute the feature counts $\mathbf{R}$ would observe in expectation from the true $\theta$ and then select actions that maximize similarity to these target features. If $\phi_\theta$ are the expected feature counts induced by $\theta$ then this scheme amounts to the following decision rule:

$$\tau^{\mathbf{H}} \leftarrow \underset{\tau}{\mathrm{argmax}} \ \phi(\tau)^{\top}\theta - \eta||\phi_\theta - \phi(\tau)||^2. \tag{1}$$

This rule selects a trajectory that trades off between the sum of rewards $\phi(\tau)^{\top}\theta$ and the feature dissimilarity $||\phi_\theta - \phi(\tau)||^2$. Note that this is generally distinct from the action selected by the demonstration-by-expert policy. The goal is to match the expected sum of features under a *distribution* of trajectories with the sum of features from a *single* trajectory. The correct measure of feature

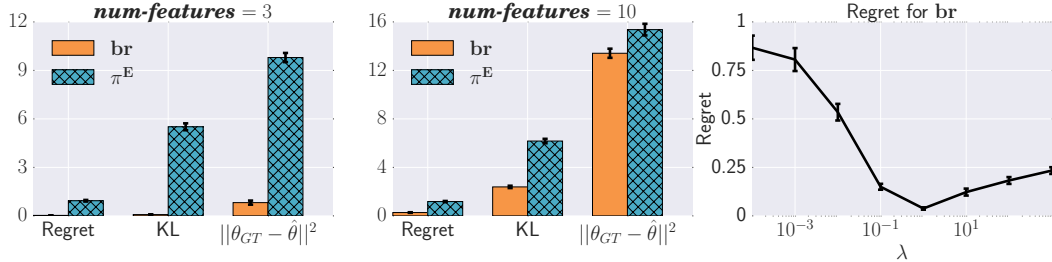

Figure 2: Left, Middle: Comparison of 'expert' demonstration ($\pi^{\mathbf{E}}$) with 'instructive' demonstration (**br**). Lower numbers are better. Using the best response causes **R** to infer a better distribution over $\theta$ so it does a better job of maximizing reward. Right: The regret of the instructive demonstration policy as a function of how optimal **R** expects **H** to be. $\lambda = 0$ corresponds to a robot that expects purely random behavior and $\lambda = \infty$ corresponds to a robot that expects optimal behavior. Regret is minimized for an intermediate value of $\lambda$: if $\lambda$ is too small, then **R** learns nothing from its observations; if $\lambda$ is too large, then **R** expects many values of $\theta$ to lead to the same trajectory so **H** has no way to differentiate those reward functions.

similarity is *regret*: the difference between the reward **R** would collect if it knew the true $\theta$ and the reward **R** actually collects using the inferred $\theta$. Computing this similarity is expensive, so we use an $\ell_2$ norm as a proxy measure of similarity.

# 4 Experiments

## 4.1 Cooperative Learning for Mobile Robot Navigation

Our experimental domain is a 2D navigation problem on a discrete grid. In the learning phase of the game, **H** teleoperates a trajectory while **R** observes. In the deployment phase, **R** is placed in a random state and given control of the robot. We use a finite horizon $H$, and let the first $\frac{H}{2}$ timesteps be the learning phase. There are $N_\phi$ state features defined as radial basis functions where the centers are common knowledge. Rewards are linear in these features and $\theta$. The initial world state is in the middle of the map. We use a uniform distribution on $[-1, 1]^{N_\phi}$ for the prior on $\theta$. Actions move in one of the four cardinal directions $\{N, S, E, W\}$ and there is an additional no-op $\emptyset$ that each actor executes deterministically on the other agent's turn.

Figure 1 shows an example comparison between demonstration-by-expert and the approximate best response policy in Section 3.4. The leftmost image is the ground truth reward function. Next to it are demonstration trajectories produce by these two policies. Each path is superimposed on the maximum a-posteriori reward function the robot infers from the demonstration. We can see that the demonstration-by-expert policy immediately goes to the highest reward and stays there. In contrast, the best response policy moves to both areas of high reward. The robot reward function the robot infers from the best response demonstration is much more representative of the true reward function, when compared with the reward function it infers from demonstration-by-expert.

## 4.2 Demonstration-by-Expert vs Best Responder

**Hypothesis.** When **R** plays an IRL algorithm that matches features, **H** prefers the best response policy from Section 3.4 to $\pi^{\mathbf{E}}$: the best response policy will significantly outperform the DBE policy.

**Manipulated Variables.** Our experiment consists of 2 factors: **H-*policy*** and ***num-features***. We make the assumption that **R** uses an IRL algorithm to compute its estimate of $\theta$ during learning and maximizes reward under this estimate during deployment. We use Maximum-Entropy IRL (Ziebart et al., 2008) to implement **R**'s policy. **H-*policy*** varies **H**'s strategy $\pi^{\mathbf{H}}$ and has two levels: demonstration-by-expert ($\pi^{\mathbf{E}}$) and best-responder (**br**). In the $\pi^{\mathbf{E}}$ level **H** maximizes reward during the demonstration. In the **br** level **H** uses the approximate algorithm from Section 3.4 to compute an approximate best response to $\pi^{\mathbf{R}}$. The trade-off between reward and communication $\eta$ is set by cross-validation before the game begins. The ***num-features*** factor varies the dimensionality of $\phi$ across two levels: 3 features and 10 features. We do this to test whether and how the difference between experts and best-responders is affected by dimensionality. We use a factorial design that leads to 4 distinct conditions. We test each condition against a random sample of $N = 500$ different reward parameters. We use a within-subjects design with respect to the the **H-policy** factor so the same reward parameters are tested for $\pi^{\mathbf{E}}$ and **br**.

**Dependent Measures.** We use the regret with respect to a fully-observed setting where the robot knows the ground truth $\theta$ as a measure of performance. We let $\hat{\theta}$ be the robot's estimate of the reward parameters and let $\theta_{GT}$ be the ground truth reward parameters. The primary measure is the *regret* of **R**'s policy: the difference between the value of the policy that maximizes the inferred reward $\hat{\theta}$ and the value of the policy that maximizes the true reward $\theta_{GT}$. We also use two secondary measures. The first is the KL-divergence between the maximum-entropy trajectory distribution induced by $\hat{\theta}$ and the maximum-entropy trajectory distribution induced by $\theta$. Finally, we use the $\ell_2$-norm between the vector or rewards defined by $\hat{\theta}$ and the vector induced by $\theta_{GT}$.

**Results.** There was relatively little correlation between the measures (Cronbach's $\alpha$ of .47), so we ran a factorial repeated measures ANOVA for each measure. Across all measures, we found a significant effect for **H-*policy***, with **br** outperforming $\pi^{\mathbf{E}}$ on all measures as we hypothesized (all with $F > 962$, $p < .0001$). We did find an interaction effect with ***num-features*** for KL-divergence and the $\ell_2$-norm of the reward vector but post-hoc Tukey HSD showed **br** to always outperform $\pi^{\mathbf{E}}$. The interaction effect arises because the gap between the two levels of **H-*policy*** is larger with fewer reward parameters; we interpret this as evidence that ***num-features*** $= 3$ is an easier teaching problem for **H**. Figure 2 (Left, Middle) shows the dependent measures from our experiment.

### 4.3   Varying R's Expectations

Maximum-Entropy IRL includes a free parameter $\lambda$ that controls how optimal **R** expects **H** to behave. If $\lambda = 0$, **R** will update its belief as if **H**'s observed behavior is *independent* of her preferences $\theta$. If $\lambda = \infty$, **R** will update its belief as if **H**'s behavior is *exactly* optimal. We ran a followup experiment to determine how varying $\lambda$ changes the regret of the **br** policy.

Changing $\lambda$ changes the forward model in **R**'s belief update: the mapping **R** hypothesizes between a given reward parameter $\theta$ and the observed feature counts $\phi_\theta$. This mapping is many-to-one for extreme values of $\lambda$. $\lambda \approx 0$ means that all values of $\theta$ lead to the same expected feature counts because trajectories are chosen uniformly at random. Alternatively, $\lambda >> 0$ means that almost all probability mass falls on the optimal trajectory and many values of $\theta$ will lead to the same optimal trajectory. This suggests that it is easier for **H** to differentiate different values of $\theta$ if **R** assumes she is noisily optimal, but only up until a maximum noise level. Figure 2 plots regret as a function of $\lambda$ and supports this analysis: **H** has less regret for intermediate values of $\lambda$.

## 5   Conclusion and Future Work

In this work, we presented a game-theoretic model for cooperative learning, CIRL. Key to this model is that the robot *knows* that it is in a shared environment and is attempting to maximize the human's reward (as opposed to estimating the human's reward function and adopting it as its own). This leads to cooperative learning behavior and provides a framework in which to design HRI algorithms and analyze the incentives of both actors in a reward learning environment.

We reduced the problem of computing an optimal policy pair to solving a POMDP. This is a useful theoretical tool and can be used to design new algorithms, but it is clear that optimal policy pairs are only part of the story. In particular, when it performs a centralized computation, the reduction assumes that we can effectively program both actors to follow a set coordination policy. This is clearly infeasible in reality, although it may nonetheless be helpful in training humans to be better teachers. An important avenue for future research will be to consider the coordination problem: the process by which two independent actors arrive at policies that are mutual best responses. Returning to Wiener's warning, we believe that the best solution is not to put a specific purpose into the machine at all, but instead to design machines that provably converge to the right purpose as they go along.

### Acknowledgments

This work was supported by the DARPA Simplifying Complexity in Scientific Discovery (SIMPLEX) program, the Berkeley Deep Drive Center, the Center for Human Compatible AI, the Future of Life Institute, and the Defense Sciences Office contract N66001-15-2-4048. Dylan Hadfield-Menell is also supported by a NSF Graduate Research Fellowship.

## Footnotes

[2]A coordination problem of the type described in Boutilier (1999) arises if there are multiple optimal policy pairs; we defer this issue to future work.

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
