[Supplementary Material · supplementary.pdf]

# Cooperative Inverse Reinforcement Learning: Supplementary Material

October 27, 2016

### Abstract

This document contains supplementary material and proofs for the NIPS submission Cooperative Inverse Reinforcement Learning. Some parts of the main text are repeated for completeness.

## 1 CIRL Formulation

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

**Definition 2.** *Let $M$ be a CIRL game between **H** and **R**. The corresponding* coordination POMDP $M_{\mathbf{C}}$ *is a* POMDP *where the single actor is a coordinator **C**. States are tuples of world state and reward parameters: $\mathcal{S}_c = \mathcal{S} \times \Theta$. The intial state distribution places the same distribution on $\mathcal{S} \times \Theta$ as $P_0$. **C**'s actions are tuples $(\delta^{\mathbf{H}}, a^{\mathbf{R}})$ that specify an action for **R** and a decision rule for **H** that maps its private information ($\theta$) to an action $\delta^{\mathbf{H}} : \Theta \to \mathcal{A}^{\mathbf{H}}$. **C** observes **H**'s action and the world state. Transitions are defined analogously to those in $M$.*

**Theorem 1.** *Let $M$ be an arbitrary CIRL game with state space $\mathcal{S}$ and reward space $\Theta$. There exists a (single-actor)* POMDP $M_{\mathbf{C}}$ *with (hidden) state space $\mathcal{S}_{\mathbf{C}}$ such that*

$|\mathcal{S}_{\mathbf{C}}| = |\mathcal{S}| \cdot |\Theta|$ *and, for any policy pair in* $M$, *there is a policy in* $M_{\mathbf{C}}$ *that achieves the same sum of discounted rewards.*

*Proof.* We take $M_{\mathbf{C}}$ to be the coordination POMDP associated associated with $M$. The second component of **C**'s action is an action for **R**. **R** has no private observations, so for any policy $\pi^{\mathbf{R}}$ **R** could choose to follow, **C** can match it by simulating $\pi^{\mathbf{R}}$ and outputting the corresponding action. Similarly, **C** only observes common observations, so **R** can implement any coordinator strategy by simulating **C** and directly executing the appropriate action.

By a similar arguement, **H** can also simulate any given $\pi^{\mathbf{C}}$ to compute her decision rule $\delta^{\mathbf{H}}$, and then execute the corresponding action. To see that there is a $\pi^{\mathbf{C}}$ that can reproduce the behavior of any $\pi^{\mathbf{H}}$, let $h$ be the action-observation history for **H**. **C** can choose the following decision rule

$$\delta^{\mathbf{H}}(\theta) = \pi^{\mathbf{H}}(\theta; h)$$

to produce the same behavior. $\qquad\square$

**Corollary 1.** *Let* $M$ *be a CIRL game. There exist optimal policies* $(\pi^{\mathbf{H}^*}, \pi^{\mathbf{R}^*})$ *that only depend on the current state and* **R**'s *belief.*

$$\pi^{\mathbf{H}^*} : \mathcal{S} \times \Delta_{\Theta} \times \Theta \to \mathcal{A}^{\mathbf{H}}, \qquad \pi^{\mathbf{R}^*} : \mathcal{S} \times \Delta_{\Theta} \to \mathcal{A}^{\mathbf{R}}.$$

*Proof.* Smallwood & Sondik (1973) showed that an optimal policy in a POMDP only depends on the belief state. **R**'s belief uniquely determines the belief for **C**. From this, an appeal to Theorem 1 shows the result. $\qquad\square$

## 2 Apprenticeship CIRL

**Example.** Consider an example apprenticeship task where **R** needs to help **H** make office supplies. **H** and **R** can make paperclips and staples and the unobserved $\theta$ describe **H**'s preference for paperclips vs staples. We model the problem as an ACIRL in which the learning and deployment phase each consist of an individual action.

The world state in this problem is a tuple $(p_s, q_s, t)$ where $p_s$ and $q_s$ respectively represent the number of paperclips and staples **H** owns. $t$ is the round number. An action is a tuple $(p_a, q_a)$ that produces $p_a$ paperclips and $q_a$ staples. The human can make 2 items total: $\mathcal{A}^{\mathbf{H}} = \{(0, 2), (1, 1), (2, 0)\}$. The robot has different capabilities. It can make 50 units of each item or it can choose to make 90 of a single item: $\mathcal{A}^{\mathbf{R}} = \{(0, 90), (50, 50), (90, 0)\}$.

We let $\Theta = [0, 1]$ and define $R$ so that $\theta$ indicates the relative preference between paperclips and staples: $R(s, (p_a, q_q); \theta) = \theta p_a + (1 - \theta)q_a$. **R**'s action is ignored when $t = 0$ and **H**'s is ignored when $t = 1$. At $t = 2$, the game is over, so we transition to a sink state, $(0, 0, 2)$. Initially, there are no paperclips or staples and we use a uniform prior on $\theta$.

**H** only acts in the initial state, so $\pi^{\mathbf{H}}$ can be entirely describe by a single decision rule $\delta^{\mathbf{H}} : [0, 1] \to \mathcal{A}^{\mathbf{H}}$. **R** only observes one action from **H** and so the reachable beliefs are in one-to-one correspondence with **H**'s actions. This lets us characterize **R**'s policy as $\pi^{\mathbf{R}} : \mathcal{A}^{\mathbf{H}} \to \mathcal{A}^{\mathbf{R}}$.

**Theorem 2.** *Let $M$ be an ACIRL game. In the deployment phase, the optimal policy for* $\mathbf{R}$ *maximizes reward in the* MDP *induced by the mean* $\theta$.

*Proof.* If $\mathbf{R}$ never observes another action from $\mathbf{H}$, then there are no common observations so the coordination POMDP has no observations. The unobserved component of the state is static, so this distribution does not change over time. This reduces the problem to solving an MDP under a fixed distribution over reward functions so Theorem 3 from Ramachandran & Amir (2007) shows the result. □

The DBE assumption in our example assumes that $\mathbf{H}$ maximize reward in the first round. Let $\theta = 0.49$. $\mathbf{H}$ maximizes reward and chooses to make 0 paperclips and 2 staples. $\mathbf{R}$ observes this and updates its belief (using $\delta^{\mathbf{E}}$ to define the observation distribution). In this case, we get $b^{\mathbf{R}} = \mathbf{Unif}([0, 0.5))$. Given this belief, $\mathbf{R}$'s maximizes expected reward and chooses to make 0 paperclips and 90 staples. Thus, the expert decision rule $\delta^{\mathbf{E}}$ and its *best response* $\mathbf{br}(\delta^{\mathbf{E}})$ are defined by

$$
\delta^{\mathbf{E}}(\theta) = \begin{cases} (0,2) & \theta < 0.5 \\ (1,1) & \theta = 0.5 \\ (2,0) & \theta > 0.5 \end{cases}, \tag{1}
$$

$$
\mathbf{br}(\delta^{\mathbf{E}})(a^{\mathbf{H}}) = \begin{cases} (0,90) & a^{\mathbf{H}} = (0,2) \\ (50,50) & a^{\mathbf{H}} = (1,1) \\ (90,0) & a^{\mathbf{H}} = (2,0) \end{cases}. \tag{2}
$$

Note that when $\theta = 0.49$ $\mathbf{H}$ would prefer $\mathbf{R}$ to choose (50, 50). $\mathbf{H}$ is willing to forgo immediate reward during the demonstration to communicate this to $\mathbf{R}$: the best response chooses $(1,1)$ when $\theta = 0.49$. This leads to the following result.

**Theorem 3.** *There exist ACIRL games where the best-response for $\mathbf{H}$ to $\pi^{\mathbf{R}}$ violates the expert demonstrator assumption. In other words, if $\mathbf{br}(\pi)$ is the best response to $\pi$, then $\mathbf{br}(\mathbf{br}(\pi^{\mathbf{E}})) \neq \pi^{\mathbf{E}}$.*

*Proof.* Our office supply example gives a counter example that shows the theorem. When $\mathbf{H}$ accounts for $\mathbf{R}$'s actions under $\mathbf{br}(\delta^{\mathbf{E}})$, $\mathbf{H}$ is faced with a choice between 0 paperclips and 92 staples, 51 of each, or 92 paperclips and 0 staples. It is straightforward to show that the optimal decision rule is given by

$$
\delta^{\mathbf{H}}(\theta) = \begin{cases} (0,2) & \theta < \frac{41}{92} \\ (1,1) & \frac{41}{92} \leq \theta \leq \frac{51}{92} \\ (2,0) & \theta > \frac{51}{92} \end{cases}.
$$

This is distinct from Equation 1 so we conclude the result. □