[Reviews · NeurIPS 2016]

Reviewer 1

Summary

This paper introduces so called cooperative inverse reinforcement learning (CIRL) model for learning from demonstration problems. It assumes the learning process is a two-player Markov game with identical payoffs between a demonstrator and a learner. Authors reduce the computation of solving the Markov game as solving a POMDP problem making the model computation applicable to small problems in practice. The authors formulate the apprenticeship learning problem as a turn-based CIRL with a learning phase and deployment phase. The key contribution of this work is to show the claim that the learner’s maximizing reward response (learner’s policy in deployment phase) to the demonstration of CIRL model (learning phase) outperforms the expert demonstration assumed in the earlier work of inverse reinforcement learning. This claim is verified by a coffee supplier counter example and also a 2D navigation grid experimental setting. This work seems novel and can be useful in the case where the learner can interact with the demonstrator. However, the experiment seems not sufficient to convince us that the model is useful in practice especially since there is no real data experimental setting.

Qualitative Assessment

1.The counter example assumes different action domains of demonstrator and learner. Is this the reason that the best-response for H to \pi^R violates the expert demonstrator assumption? What if the embodiments setting of demonstrator and learner are the same? 2.It is important to have real experimental result to convince us that the CIRL works in practice. 3.Corollary 1 doesn’t interpret the symbol \Delta^\theta which I assume is the domain of R’s belief.

Confidence in this Review

2-Confident (read it all; understood it all reasonably well)


Reviewer 2

Summary

The paper presents a general framework for cooperative interaction between a human and a robot. This framework encompasses well-known settings such as inverse reinforcement learning (IRL) or optimal teaching. The authors argue that it can be cast as a POMDP problem and then present in more details how apprentice learning can be formalized in their framework. This leads to the following insight: if the human knows that the robot uses IRL, she may be better off not demonstrating with an optimal policy, but with a “best response” to the robot’s strategy. The authors also provide an approximation scheme in the case where rewards are expressed as linear combinations of state features and when the human demonstrate only one trajectory. Finally, experimental results in the navigation problem support the previous insight.

Qualitative Assessment

My opinion on this paper is quite mixed. On the one hand, the paper is generally clear. I like this general unifying framework and the insight it provides. On the other hand, the exposition could be made more precise and more rigorous. Some technical explanations and the proofs are a bit hand wavy. Besides, the theoretical results seem quite straightforward. Here are some more detailed comments: - Nash equilibrium is a concept developed in non-cooperative games. As the proposed framework is cooperative, the terminology from non-cooperative games may not be the best. - I find it strange to assume that both agents know the probability distribution over \theta. H doesn’t simply choose \theta? The model is a representation from the viewpoint of the robot and so maybe P_0 could simply be interpreted as its prior belief. - I think the authors should give the formal definition of the coordination POMDP to make things clearer. For instance, the explanations given in the paper and the supplementary material do not match: for instance, for the POMDP states (l.212, l.288 in paper and Def.2 in supplementary material). - In Sec.4.2, does the IRL approach also demonstrate only one trajectory? Assuming this is the case, could the experimental results be explained by the fact that only one trajectory is demonstrated? Some typos: l.88: Should the histories be defined by S x (A^H x A^R x S)^*? l.250: it gets l.270: gives use the ability -> gives the ability? l.272: although obvious, br is not defined l.279: Remark 3 seems not to be really related to what precedes it. l.287: a POMDP l.321: Why is Figure placed on p.3? l.326: The robot reward function -> The reward function l.407: Kand -> K, and

Confidence in this Review

2-Confident (read it all; understood it all reasonably well)


Reviewer 3

Summary

Proposes a new model called Cooperative Inverse Reinforcement Learning for modeling settings where a human and robot act and both agents try to maximize the human's reward but the robot does not observe the human's reward initially and must infer it, giving incentive to human to take actions that may sacrifice immediate reward in order to be more informative to the robot.

Qualitative Assessment

They present a novel model that seems like it could potentially have practical impact. There is theoretical and experimental evaluation. The theoretical results did not seem particularly deep, and I think the main value of the contribution rests on how realistic/important the new conceptual model is for modeling realistic scenarios. I would start with a motivating example much earlier, provide more intuition for why it is important, and describe important real-world scenarios that it exemplifies. The first example is not until page 6 line 238. It would be helpful to have more discussion for why this game is interesting (beyond just the description of it and analysis), and how modeling it in just IRL setting would have changed the analysis. I think more discussion and intuition could also be provided for the 313-320 example. It isn't really clear to me what is happening: the human is trying to navigate a grid and the robot is trying to help it? The performance improvement of using the new best-response approach over the prior expert demonstration approach for this example seems quite significant. The discussion of prior and related work is very comprehensive. King Midas analogy could be elaborated on, since not all readers might be familiar. They mention the possibility of multiple optimal policies in footnote 1 and say they defer it to future work, but I'd like to see a little more discussion. Two-player general sum games can have multiple Nash equilibia (each with potentially different values to the players), and I'd like at least some justification for why it is the right solution concept for this setting. In 4.1 I'd avoid using bold H for the human agent and non-bold H for time horizon (T may be better).

Confidence in this Review

2-Confident (read it all; understood it all reasonably well)


Reviewer 4

Summary

The paper proposes a game model for cooperative learning in Human-Robot scenario. It formalises an inverse RL problem within the framework of cooperative partial information two-player game.

Qualitative Assessment

The paper deals with interesting problem cooperative learning in Human-Robot setup. The author has shown how their approach relates to the current SoA. The paper is well-written; however the overall contribution (both technical quality and novelty) is doubtful. The main arguments supporting this statement are: • The DBE assumption is strong enough and I doubt whether it can be fully applied to practical examples. Reason: strict dependence of H’s reward on success of R’s learning (H’s internal reward function) has a significant influence on R’s learning (a kind of positive feedback). • It is unclear what cooperation is supposed in the described H-R scenario when H and R collectively solve a single decision problem and how the considered type of cooperation influences the reward function. This should be commented. • Proof of Theorem 1 provided in the supplementary material is either wrong or not clear enough. In particular statement “R can also simulate C” contradicts Definition 2, in particular definition of C’s actions, which include a decision rule for H. • Proof of Theorem 3 provided in the supplementary material is does not sound mathematically. • Feature function and its role is not clearly described. • Minor: Some symbols and notions are not introduced: for instance \delta_{\Theta} is explained neither in the main text (p.2 line 217), nor in supplementary material (p.2, Corollary 1); \eta (page 7, line 299) is not introduced too.

Confidence in this Review

2-Confident (read it all; understood it all reasonably well)